# Building models of topological quantum criticality from pivot Hamiltonians

Nathanan Tantivasadakarn[1,2], Ryan Thorngren[2,3,4,5],
Ashvin Vishwanath[2] and Ruben Verresen[2]

**1** Walter Burke Institute for Theoretical Physics and Department of Physics,
California Institute of Technology, Pasadena, CA 91125, USA
**2** Department of Physics, Harvard University, Cambridge, MA 02138, USA
**3** Kavli Institute of Theoretical Physics, University of California,
Santa Barbara, California 93106, USA
**4** Center of Mathematical Sciences and Applications,
Harvard University, Cambridge, MA 02138, USA
**5** Department of Physics, Massachusetts Institute of Technology,
Cambridge, MA 02139, USA

## Abstract

Progress in understanding symmetry-protected topological (SPT) phases has been greatly aided by our ability to construct lattice models realizing these states. In contrast, a systematic approach to constructing models that realize quantum critical points between SPT phases is lacking, particularly in dimension $d > 1$. Here, we show how the recently introduced notion of the pivot Hamiltonian—generating rotations between SPT phases—facilitates such a construction. We demonstrate this approach by constructing a spin model on the triangular lattice, which is midway between a trivial and SPT phase. The pivot Hamiltonian generates a $U(1)$ pivot symmetry which helps to stabilize a direct SPT transition. The sign-problem free nature of the model—with an additional Ising interaction preserving the pivot symmetry—allows us to obtain the phase diagram using quantum Monte Carlo simulations. We find evidence for a direct transition between trivial and SPT phases that is consistent with a deconfined quantum critical point with emergent $SO(5)$ symmetry. The known anomaly of the latter is made possible by the non-local nature of the $U(1)$ pivot symmetry. Interestingly, the pivot Hamiltonian generating this symmetry is nothing other than the staggered Baxter-Wu three-spin interaction. This work illustrates the importance of $U(1)$ pivot symmetries and proposes how to generally construct sign-problem-free lattice models of SPT transitions with such anomalous symmetry groups for other lattices and dimensions.


---

# 1 Introduction

Within the Landau paradigm, phases of matter are distinguished by different patterns of symmetry breaking. Order parameters, which characterize the pattern, form the basis of the highly successful Landau-Ginzburg-Wilson (LGW) theory of transitions between such phases. More recently, it has become well-established that there are a multitude of quantum phases [1] and even transitions between certain ordered phases—'deconfined' quantum critical points—going beyond this framework [2]. The search for new realizations of quantum phase transitions is a golden opportunity to discover new forms of universal phenomena outside the LGW paradigm [3]. A family of strongly interacting quantum phases that are well understood are the symmetry protected topological (SPT) states of bosons or spin systems, that include examples in 1D such as the Haldane-Affleck-Kennedy-Lieb-Tasaki spin-1 Heisenberg chain [4, 5] and related states [6–9], as well as extensions in general dimensions [10] including bosonic analogs of integer quantum Hall states [11, 12] and SPT phases which are stabilized by just an Ising symmetry [13, 14] in 2+1D, as well as 3D extensions [10, 15] that are bosonic versions of topological insulators and superconductors. Such a rich landscape of phases also points to a wealth of interesting quantum critical points separating distinct phases. In fact, such quantum criticality between SPT phases is intimately related to the anomalous surface states in one higher dimension [15–17] and are therefore reflected in the properties of the phases themselves. Deep connections between such topological phase transitions and deconfined quantum critical points, as well as self duality, have also been explored [15, 18–35]. Other theoretical developments include a study of 1+1D quantum phase transitions between SPT phases, 2D topological transition between bosonic 'integer' quantum Hall states and other interacting SPTs [7, 20, 36–48], and symmetry-protected quantum criticality [49–54].

However, the number of unbiased studies of transitions between SPT phases and topolog-

ical orders in lattice models beyond $d = 1$ are relatively few. The reason for this is twofold. Firstly, numerical approaches that can handle the large system sizes needed for exploring quantum criticality require a sign-problem-free realization. Indeed, certain phases are known to have an intrinsic sign problem [55–58]. Secondly, in the absence of large continuous symmetry groups, direct continuous transitions are often interrupted by direct first order transitions or intervened by intermediate phases [59–61] (which can nevertheless be exotic such as the gapless stripe phase reported in Ref. [61]). Essentially, there is a lack of concrete systematic tools for generating viable lattice models for studying topological criticality.

Here, we will show that pivot Hamiltonians—introduced in a companion work [62]—provide a new tool to attack this problem. As we will discuss in more detail, a pivot Hamiltonian generates a unitary circuit that maps a trivial phase to a given non-trivial SPT phase. Remarkably, these pivot Hamiltonians can sometimes generate $U(1)$ symmetries upon tuning between the trivial and SPT phase. In the present work, we demonstrate how this additional structure can aid the search for lattice models with direct SPT transitions, even for SPT phases protected by discrete symmetries. In particular, it leads to a prescription for constructing sign-problem-free lattice models with the anomalous symmetry group necessary to describe SPT transitions [17, 44, 63].

We now summarize the main results of this work. In the first half, we focus on a case study on the triangular lattice, where we can naturally define a $\mathbb{Z}_2^3$ symmetry associated to spin-flips on each of the three sublattices, labeled $A, B, C$ (see Fig. 1):

$$P_A = \prod_{v \in A} X_v, \qquad P_B = \prod_{v \in B} X_v, \qquad P_C = \prod_{v \in C} X_v, \qquad (1)$$

with $X, Y, Z$ denoting the Pauli matrices. It is known [64] that the following pivot Hamiltonian generates an SPT-entangler for this $\mathbb{Z}_2^3$ symmetry:

$$H_{\text{pivot}} = \frac{1}{8} \sum_{a,b,c \in \Delta} Z_a Z_b Z_c - \frac{1}{8} \sum_{a,b,c \in \nabla} Z_a Z_b Z_c, \qquad (2)$$

where we sum over all triangles of the lattice, with the sign differing for up- and down-pointing triangles. More precisely, evolving the trivial $H_0 = -\sum_v X_v$ under a $\pi$-rotation, we obtain:

$$H_{\text{SPT}} = e^{-i\pi H_{\text{pivot}}} H_0 e^{i\pi H_{\text{pivot}}} = -\sum_v \; \vcenter{\hbox{\includegraphics{hexagon}}} \;, \qquad (3)$$

where each blue line connecting two vertices denotes a Controlled-$Z$ gate (i.e., $H_{\text{SPT}}$ is a seven-site Hamiltonian). Note that the pivot Hamiltonian (2) is simply the Baxter-Wu model [65] with a staggered sign; in the absence of this staggering, we would find that it does not generate a $\mathbb{Z}_2^3$ symmetric model after a $\pi$-rotation. Furthermore, the ground state of $H_{\text{SPT}}$ is the hypergraph state [66] on the triangular lattice, since $e^{-i\pi H_{\text{pivot}}} = \prod_{\Delta, \nabla} CCZ$, where $CCZ$ denotes the Controlled-controlled-$Z$ gate.

While it is obvious that at $H_0 + H_{\text{SPT}}$ (and only at this point in the interpolation between these two Hamiltonians) has a $\mathbb{Z}_2$ symmetry generated by this SPT-entangler $e^{-i\pi H_{\text{pivot}}}$ (which indeed squares to unity), a general theorem proven in our companion paper [62] for a large class of models (including this one) shows that this is in fact enhanced to a full $U(1)$ symmetry:

$$[H_0 + H_{\text{SPT}}, H_{\text{pivot}}] = 0. \qquad (4)$$

A particularly interesting property of such a $U(1)$ pivot symmetry is that it shares a mutual anomaly with the symmetry protecting the SPT phase, in this case $\mathbb{Z}_2^3$. An anomaly means that

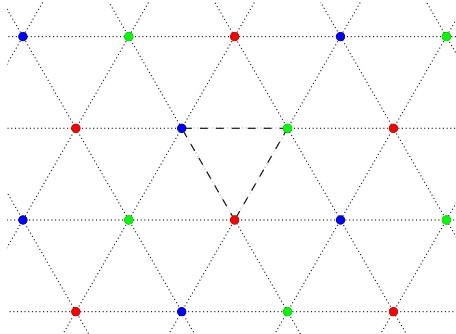

Figure 1: The triangular lattice. Qubits are placed on the vertices, which are colored in red, green and blue. The $\mathbb{Z}_2^3$ symmetry is generated by spin flips on each of the individual colors (which we label, the A,B,C sublattices). The pivot is an alternating sum of a three-body Ising interaction $ZZZ$ over all triangles.

there exist no gapped symmetric phases, giving valuable information about the phase diagram. Indeed, it has been well-appreciated that SPT phase transitions can give rise to an anomalous symmetry, although this is usually for a *discrete* duality symmetry [17, 44, 63]. Here, this discrete symmetry is enhanced to a full $U(1)$. This also explains why the pivot in Eq. (2) has to be non-onsite, since a fully onsite symmetry group cannot be anomalous. Hence, although non-onsite $U(1)$ generators have rarely been explicitly pointed out in lattice models, we stress that they can be common occurrences at SPT transitions.

This combination of the symmetry being continuous and anomalous should likely help to locate topological criticality, even for SPT phases protected by discrete symmetry. That being said, the interpolation $H_0 + H_{\text{SPT}}$ does not turn out to automatically give rise to a direct transition: Ref. [60] recently studied this model and found an intermediate ferromagnetic phase. In the present work, our motivation is to penalize the ferromagnetic phase with a term that preserves the $U(1)$ pivot symmetry, in search of topological (multi-)criticality. A natural perturbation fitting these criteria is a nearest-neighbor antiferromagnetic Ising interaction within each of the three sublattices:

$$H_{\text{Ising}} = \sum_{\Lambda=A,B,C} \sum_{\langle v_1, v_2 \rangle \in \Lambda} Z_{v_1} Z_{v_2}. \tag{5}$$

In summary, the complete model we study is:

$$H(\alpha, J) = (1-\alpha)H_0 + \alpha H_{\text{SPT}} + J H_{\text{Ising}}. \tag{6}$$

Using quantum Monte Carlo methods, we obtain the phase diagram as shown in Fig. 2. We see that the FM is indeed eventually pinched off for $\alpha = 0.5$ and $J \approx 0.21$. For larger $J$, we find a direct first order transition between the SPT phases (although we will see it is unusual since it corresponds to a superfluid for the $U(1)$ pivot symmetry). We argue that, remarkably, these two regimes are separated by a deconfined quantum critical point (DQCP), which we will support through numerics and field theory arguments. In fact, its universality is that of the $SO(5)$ DQCP, which has been studied before [2, 67–77], e.g., as a direct transition between an $SO(3)$ Néel state (the analogue of the blue shaded region in Fig. 2) and a crystalline-symmetry-breaking valence bond solid (analogous to the blue dashed line).

While the above case study concerns a scenario where the $U(1)$ pivot appears from a direct interpolation between a paramagnet and an SPT model, in the second part of this work we discuss a symmetrization procedure for obtaining lattice models with an onsite symmetry $G$ and a $U(1)$ pivot symmetry. We expect a similar phenomenology for such anomalous models as in our case study, which can be further explored in future work.

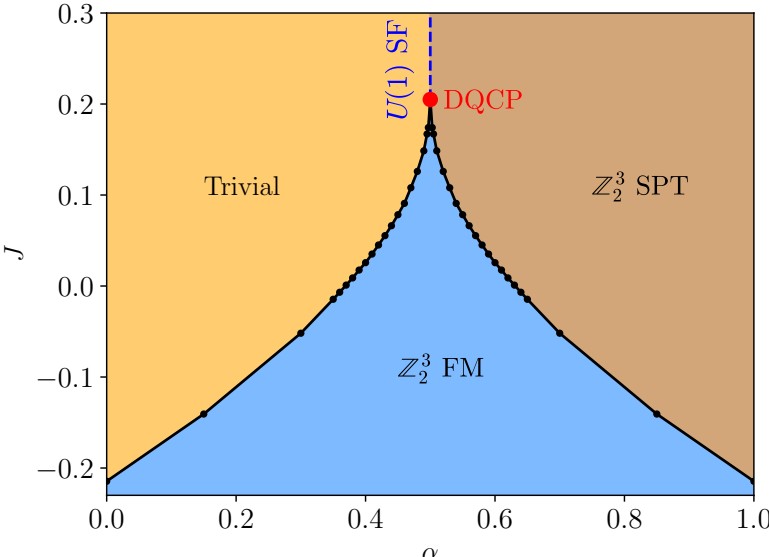

Figure 2: Phase diagram of the $\mathbb{Z}_2^3$ SPT model on the triangular lattice (6), perturbed by a trivial paramagnet (horizontal direction) and a same-sublattice Ising coupling (vertical direction). The central vertical axis has an exact $U(1)$ pivot symmetry generated by a three-site interaction (2), which has a mutual anomaly with the $\mathbb{Z}_2^3$ symmetry (1). Each black dot along the phase boundary was obtained by determining the Binder ratio crossing of the ferromagnetic (FM) order parameter. The trivial and SPT phases are separated by either an intermediate FM phase (blue shaded region) or a first order transition (which is a superfluid (SF) for the $U(1)$ pivot symmetry). We argue that these two phases meet at a multicritical point which is described by the $SO(5)$ DQCP (red dot), where $\alpha$ and $J$ correspond to perturbing with the monopole and mass operators, respectively. The $J = 0$ line was studied in Ref. [60].

The remainder of this paper is structured as follows: Sec. 2 discusses the phase diagram and the corresponding numerical results supporting the DQCP. In Sec. 3, we review the field theory description of the DQCP and argue that the allowed relevant perturbations indeed reproduce the key characteristics of our phase diagram. In Sec. 4, we discuss generalizations for how to construct SPT transitions with $U(1)$ pivot symmetries, including 3D proposals. We conclude with prospects for future studies in Sec. 5.

## 2 Numerical Study

We study the phase diagram for parameters $\alpha \in [0, 0.5]$ using Stochastic Series Expansion QMC [78, 79] on an $L \times L$ lattice, where $L = 6, 9, 12, 15$. Note that the results for $\alpha \in [0.5, 1]$, although not sign-problem-free, can be obtained by applying the action of the $\mathbb{Z}_2$ pivot:

$$UH(\alpha, J)U^\dagger = H(1 - \alpha, J), \quad \text{with } U = e^{-i\pi H_{\text{pivot}}}. \tag{7}$$

Because of the presence of an Ising term in the Hamiltonian in the $Z$ basis, the simulation can be greatly sped up by the use of non-local updates. We develop a variant of the cluster update [80] to simulate the Hamiltonian efficiently. Our algorithm reduces to the usual cluster update for $\alpha = 0$ (which corresponds to the transverse-field Ising model on each triangular sublattice). Details of the algorithm are presented in Appendix A.

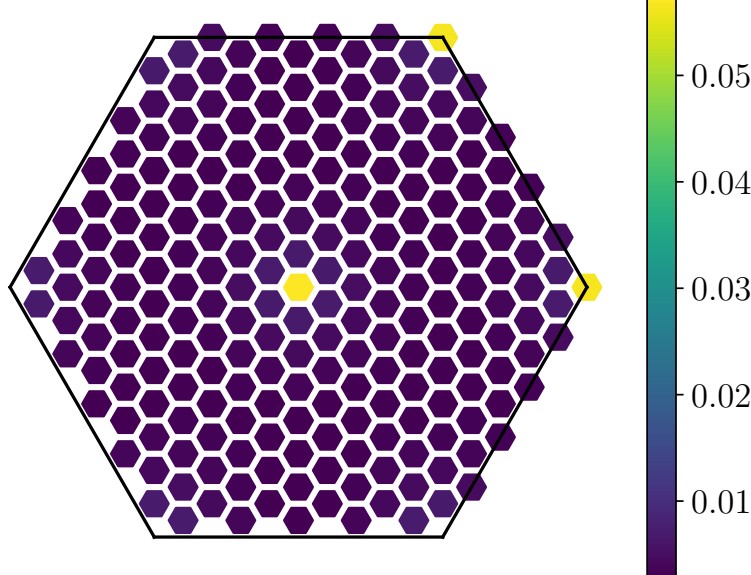

Figure 3: The structure factor for $H_0 + H_{\mathrm{SPT}}$ on a $15 \times 15$ triangular lattice (Eq. (6) with $\alpha = 0.5$ and $J = 0$). The peak of the structure factor at the center and corners of the Brillouin Zone corresponds to the pattern of a $\mathbb{Z}_2^3$ ferromagnet due to long-range ordering on each of the three sublattices of the triangular lattice.

We remark that our results for $\alpha = \frac{1}{2}$ are only consistent with the results obtained for $\alpha < \frac{1}{2}$ if we initialize in the sector where $H_{\mathrm{pivot}} = 0$. In fact, this can be seen from the following argument. First, note that $H_{\mathrm{pivot}}$ changes sign under a reflection along say, the $x$ axis, which is a symmetry of the Hamiltonian (6). Furthermore, we find from our numerics that the reflection symmetry is not spontaneously broken for the region we study. Therefore, we conclude that along the line $\alpha = \frac{1}{2}$, $H_{\mathrm{pivot}} = 0$ for the values of $J$ shown in the phase diagram. In addition, we observe that the update scheme is unable to toggle between different $U(1)$ symmetry sectors at $\alpha = 0.5$, making it essential to initialize our simulation in the correct sector. This gives an example where checking for non-onsite $U(1)$ symmetries is of actual practical significance.

Let us now systematically go through the phase diagram Fig. 2. First, the trivial and SPT phases are understood in the exactly-solvable limits in Eq. (3). The existence of the ferromagnetic (FM) phase breaking $\mathbb{Z}_2^3$ symmetry along the $\alpha = 0$ line for $J \ll 0$ is also apparent, since the model decouples to three Ising models on each triangular sublattice. This is known to have a direct transition at $J_c \approx -0.21$ [81].

The same FM phase was also observed at $J = 0$ in the range $\alpha \in [0.38, 0.62]$ in Ref. [60]. We indeed find that these two instances are in fact part of one big ferromagnetic phase corresponding to the blue shaded region in Fig. 2. The characteristic of this phase is revealed by plotting the structure factor

$$S(k) = \frac{1}{L^2} \sum_v e^{-ik(r_0 - r_v)} \langle Z_0 Z_v \rangle \,, \tag{8}$$

where the vertex 0 is some fixed vertex in the lattice. Note that the normalization is defined such that the structure factor does not diverge with system size. We plot an example for $\alpha = 0.5$ and $J = 0$ in Fig. 3, where we see a clear peak at the center and corners of the first Brillouin zone of the triangular lattice, consistent with the symmetry breaking pattern of a ferromagnet (FM) in each of the three sublattices. This agrees with the findings of Ref. [60].

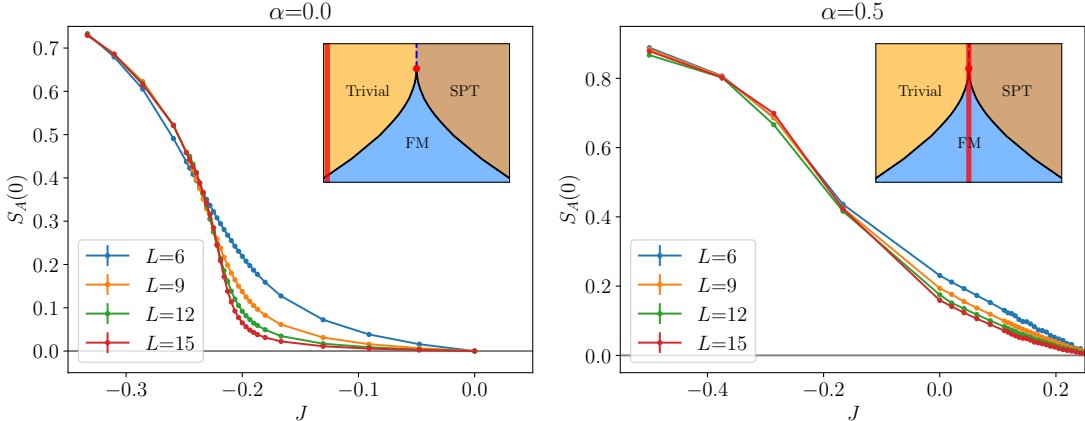

Figure 4: The structure factor at $k = 0$ for the A sublattice of the $L \times L$ triangular lattice, which is an order parameter for the ferromagnetic phase ($S_A(0)$). A clear transition is seen for $\alpha = 0$, corresponding to Ising[3] criticality. In contrast, the order parameter decays very slowly for $\alpha = 0.5$, meaning that it has a small value over a wider region of parameter space, making it more challenging to precisely locate the critical point. This region with a small magnetic moment is consistent with the phase boundary in Fig. 2 displaying a narrow ordered region upon approaching the multicritical point. In fact, the latter seems to end in a cusp, which is expected from $SO(5)$ deconfined criticality (see also Fig. 7).

We can use the structure factor as an order parameter to obtain insight into the phase boundaries between the FM and trivial phase. In Fig. 4, we plot the structure factor of the $A$ sublattice at $k = 0$ defined as[1]

$$S_A(0) = \frac{3}{L^2} \sum_{v \in A} \langle Z_0 Z_v \rangle , \tag{9}$$

where the vertex 0 now is some fixed vertex in the $A$ sublattice.

In the limit of the Ising model ($\alpha = 0$), the transition is easy to see in Fig 4. In contrast, along the self-dual line ($\alpha = 0.5$), we observe that the FM order parameter decays very slowly. This is consistent with the singular cusp behavior in our phase diagram, making a precise determination of the critical point more challenging.

To determine the critical value $J_c$ (for a given $\alpha$) more quantitatively, we use the following Binder ratio of the order parameter:

$$B = \frac{\langle m^4 \rangle}{\langle m^2 \rangle^2} . \tag{10}$$

Here, we follow Ref. [60] in that we take the order parameter of the $\mathbb{Z}_2^3$ ferromagnet to be

$$m = m_A m_B m_C , \tag{11}$$

where $m_A$ is the magnetization of the $A$ sublattice and similarly for $m_B$ and $m_C$, i.e.,

$$m_{A,B,C} = \frac{3}{L^2} \sum_{v \in A,B,C} Z_v . \tag{12}$$

---

[1]Numerically we find that looking at a a single sublattice rather than the whole $S(k = 0)$ is more stable for $J \ll 0$.

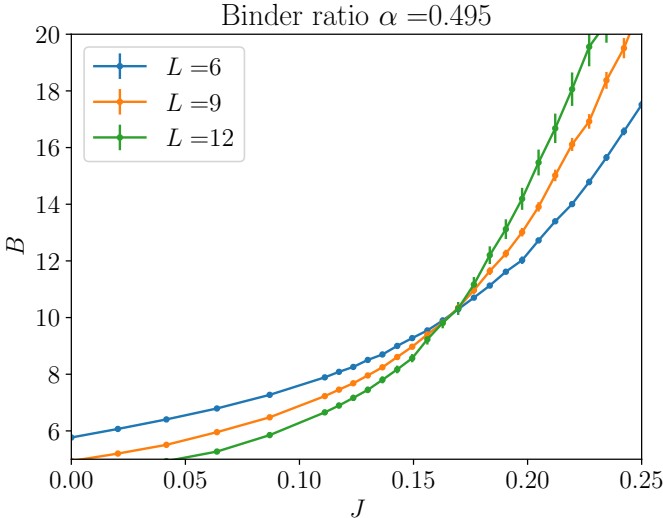

Figure 5: Binder ratio (10) at $\alpha = 0.495$. The crossing for different system sizes signifies a second order transition (which is expected to be in $O(3)$ universality with cubic anisotropy), and the critical value $J_c$ is extracted. The black dots in Fig. 2 are obtained in this way.

An example of this Binder ratio is shown in Fig. 5 for $\alpha = 0.495$, where the intersection of the Binder ratio for various system sizes signifies a continuous transition. Indeed, Ref. [60] has pointed out that this is in the $O(3)$ universality class with a cubic anisotropy [82] (henceforth, cubic anisotropy is implicitly assumed when referring to the $O(3)$). This numerical data suggests that the transition remains continuous as we approach $\alpha = 0.5$.

We claim that at $\alpha = 0.5$, the FM has a direct continuous transition to a $U(1)$ superfluid (SF). Before discussing this direct transition, let us first study this claimed SF. This is an intriguing SF for two reasons. Firstly, it spontaneously breaks the non-onsite $U(1)$ pivot symmetry given by Eq. (2). Secondly, the SF can be interpreted as a first-order transition between the trivial and SPT phases upon tuning away from $\alpha = 0.5$; indeed, this explicitly breaks the $U(1)$ pivot. It follows that this first order transition is infinitely degenerate. To see that there is indeed a SF which spontaneously breaks the $U(1)$ pivot, we can consider the order parameter $H_{\mathrm{SPT}} - H_0$, which has unit charge under the pivot. Note that from the Hellmann-Feynman theorem, the ground state energy[2] satisfies

$$\frac{dE}{d\alpha} = \left\langle \frac{dH}{d\alpha} \right\rangle = \langle H_{\mathrm{SPT}} - H_0 \rangle \,. \tag{13}$$

Therefore, the $U(1)$ SF region can be identified by a kink in the energy as a function of $\alpha$ (indeed, this precisely denotes the first-order transition between the trivial and SPT phase). In Fig. 6 we estimate the slope by fitting the energy as a function of $\alpha$ with a cubic polynomial in the range $0.47 \leq \alpha < 0.5$. For instance, we observe non-vanishing slope $J = 0.25$ compared to $J = 0$, signifying that the former is in the SF.

Having shown the existence of the $\mathbb{Z}_2^3$ FM and the $U(1)$ SF, we will now argue that there is a direct transition between these two regions. In particular, we will exclude the possibilities of an intermediate trivial phase, or an intermediate regime where both FM and SF overlap ("FM+SF"). The former is forbidden by the mutual anomaly between the $\mathbb{Z}_2^3$ and $U(1)$ symmetries (see the discussion in the introduction or in Sec. 3). If the latter were the case, then

---

[2]We have chosen a low enough temperature $T$ such that we can effectively obtain the ground state energy.

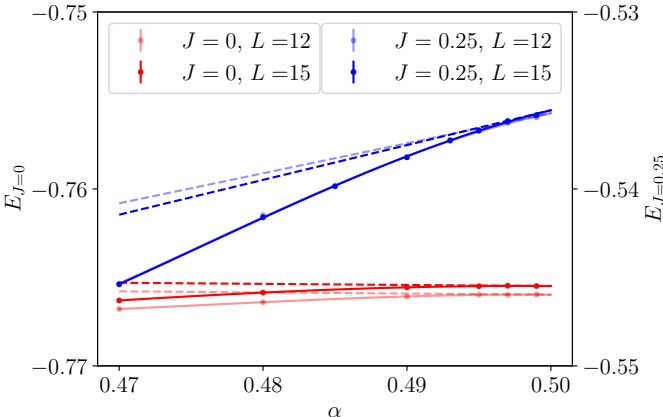

Figure 6: Due to the duality (7) which takes $\alpha \to 1-\alpha$, a nonzero slope $\frac{dE}{da}$ at $\alpha = 0.5$ signals a first order transition between the trivial and SPT phases. Moreover, due to the Hellmann-Feynman theorem (13), the slope also measures the order parameter for the $U(1)$ superfluid which spontaneously breaks the pivot (2). We see that the slope (dashed lines) obtained by fitting the data with a cubic polynomial (solid lines) is non-vanishing for $J = 0.25$, while it is approximately flat for $J = 0$ (both are shown with the same scale for comparison).

we would have a transition between FM+SF and SF. A first order transition would be inconsistent with the $O(3)$ critical line remaining continuous as we approach $\alpha \to 0.5$ (and we have already argued in favor of it remaining continuous, e.g., see Fig. 5). On the other hand, a continuous transition would have to also be in the $O(3)$ universality class. However, this is inconsistent with the cusp of the phase boundary seen in the phase diagram.[3] In fact, we find that the phase boundary $J_c(\alpha)$ forms a nice scaling upon approaching $\alpha = 0.5$ (see Fig. 7):

$$J_c(\alpha) \approx J_c(0.5) - |0.5 - \alpha|^b \,, \tag{14}$$

where the best fit value for the scaling is $b \approx 0.5$.

The above scaling formula for $J_c(\alpha)$ moreover indicates that this direct transition is continuous. Indeed, the only known mechanism for forcing a scaling law on $J_c(\alpha)$ upon approaching $\alpha \to 0.5$ is for there to be a critical point at $\alpha = 0.5$. The exponent of this scaling law encodes universal data of this limiting critical point. More precisely, the system can be said to behave as if it exhibits a continuous transition; we cannot make definite claims about the thermodynamic limit. Indeed the $SO(5)$ DQCP has been proposed to realize a 'walking criticality' scenario [72, 83–87], as suggested by bootstrap calculations [88, 89] and numerics [69, 74, 86, 90–92], although stable critical exponents have also been reported [93, 94].

In the particular case of $SO(5)$ DQCP, this scaling is related to the ratio of the scaling dimensions of the monopole operator $[z]$ (tuned by $\alpha$) to the mass operator $[z^2]$ (tuned by $J$) ($z$ denotes the complex scalar in the field theory description of the DQCP in Sec. 3)

$$b = \frac{[z]}{[z^2]} \,. \tag{15}$$

Note that these scaling dimensions are related to the critical exponents via

$$[z] = (1 + \eta)/2, \qquad\qquad [z^2] = 3 - 1/\nu \,. \tag{16}$$

---

[3]This is in the plausible assumption that the coupling of the Goldstone mode to the $O(3)$ criticality does not drastically alter the latter.

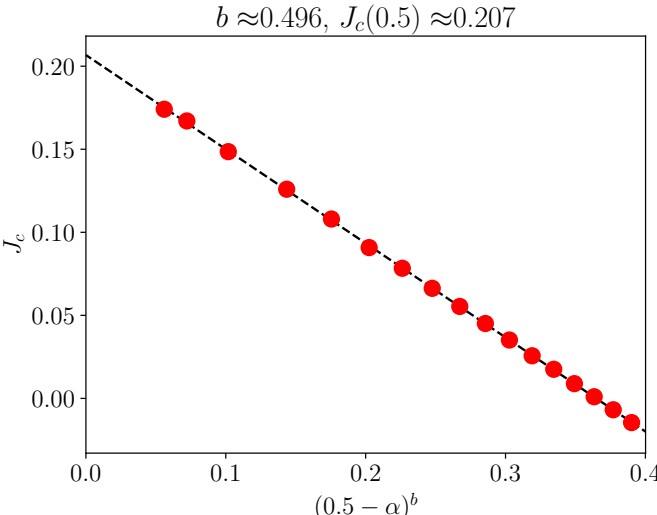

Figure 7: The phase boundary between trivial and FM phases extracted from the Binder ratio crossing (as shown in Fig. 5 for $\alpha = 0.495$) fits the scaling formula in Eq. (14). Here, the values of $\alpha$ used are in the range $[0.35, 0.497]$. This scaling is used to extrapolate the value of $J_c$ at the DQCP to be $J_c(0.5) \approx 0.21$.

Using estimates of the critical exponents for the DQCP in Ref. [86], we find that $b$ is expected to fall in the range 0.45 to 0.76, which is satisfied by our estimate in Eq. (14). As a separate sanity check, we have also numerically calculated the derivative of the energy along the $\alpha = 0.5$ line (see Fig. 10 in Appendix B) and we observe no discernible jump, consistent with the claim of a continuous transition.

Finally, we mention that the Binder-ratio can in principle be used to extract the critical exponent $\nu$ by finite size scaling. Along the $O(3)$ criticality, our extracted values of $\nu$ (for distinct $\alpha$) are consistent with field theory expectations ($\nu \approx 0.7$ for both with or without cubic anisotropy) [82,95–97] but increases rapidly as we approach the DQCP. (Relevant information on the details can be found in Appendix B.) The latter is likely an expected finite-size effect. Indeed, it is known that the critical exponents of $SO(5)$ DQCP show strong finite-size dependence, which has been demonstrated very clearly in Ref. [86] which was able to access *linear* system sizes up to $L = 512$. We should not expect to be able to extract reliable critical exponents for the available system sizes of this model (although for an alternative description of our critical model which might allow for bigger system sizes, see Sec. 4.1.1 and the discussion in the outlook).

Table 1: Correspondence between lattice and continuum symmetries. We consider the $\mathbb{Z}_2^3$ SPT model on the triangular lattice as in Eq. (6). The vicinity of the multicritical point (red dot in Fig. 2) is described by $SO(5)$ deconfined quantum criticality. Here, $R_i \in SO(5)$ (for $i = 1, 2, 3, 4$) are defined as diagonal matrices with $-1$ in the $i^{\text{th}}$ and $5^{\text{th}}$ positions.

| Group | Lattice | $O(5)$ |
|---|---|---|
| $\mathbb{Z}_2^3$ | $\prod_A X_v, \prod_B X_v, \prod_C X_v$ | $R_1, R_2, R_3$ |
| $U(1)$ | $\sum_\Delta (-1)^\Delta ZZZ$ | lower right $2 \times 2$ rotation block |
| $D_6$ | Dihedral symmetry of triangle | upper left $3 \times 3$ permutation block |

# 3 Field theory describing the DQCP

From the numerical study of the phase diagram, we can infer the phases of the model (i.e., the trivial, SPT, FM and SF regions). Moreover, we argued in favor of a direct continuous transition, corresponding to the red dot in Fig. 2. To argue that the latter is described by $SO(5)$ DQCP, it is key to relate the symmetries of our lattice model to the field-theoretic description of this universality class. Using this information, one can argue that the allowed relevant perturbations reproduce the phase diagram observed in Fig. 2. Moreover, we can match the necessary anomalies.

Let us first review the field theory description. The $\mathbb{CP}^1$ model (in the absence of magnetic monopoles [18]) can be expressed in terms of a $U(1)$ gauge field $a_\mu$ and two charge 1 complex scalars $z_1, z_2$, with a Lagrangian

$$\sum_j |(\partial_\mu - i a_\mu) z_j|^2 + m^2 \sum_j |z_j|^2 + u (\sum_j |z_j|^2)^2.$$

The scalars $z_1, z_2$ transform as a $U(2)$ fundamental, which is reduced to $SO(3) = U(2)/U(1)$ when we quotient out the gauge transformations. The $SO(3)$ vector is $n^a = z^\dagger \tau^a z$, where $\tau^a$ are Pauli matrices. There is also a $U(1)$ magnetic symmetry, whose conserved charge is nothing but the total magnetic flux of $a$, with the topologically conserved current $da$. Finally, we have charge conjugation $C$ which acts as $z \mapsto z^*$ (negating the $\tau^y$ component of $n^a$) and $a \mapsto -a$. The symmetry group can be written as

$$G = (SO(3) \times U(1)) \rtimes \mathbb{Z}_2^C. \tag{17}$$

This embeds in the larger group $SO(5)$ [72] (a proposed symmetry enhancement at a critical value of $m^2/u$) with $SO(3)$ and $U(1)$ as the upper $3 \times 3$ and lower $2 \times 2$ blocks, respectively, and $C$ as a diagonal matrix with a $-1$ in positions 2 and 5 and $+1$ in other positions, such that the $SO(5)$ vector is $(n^x, n^y, n^z, M_1 + M_{-1}, i(M_1 - M_{-1}))$, where $M_{\pm 1}$ are the monopole operators of $a$ of charge $\pm 1$. The anomaly of the $SO(5)$ symmetry can be concisely expressed as the Euler class of the vector representation, $e(V_5) \in H^5(BSO(5), \mathbb{Z})$ [72,98].

Consider the $\mathbb{Z}_2^3$ subgroup of $G$ generated in $SO(5)$ by the three diagonal matrices $R_1 = \mathrm{diag}(-1, 1, 1, 1, -1)$, $R_2 = \mathrm{diag}(1, -1, 1, 1, -1)$, and $R_3 = \mathrm{diag}(1, 1, -1, 1, -1)$. Note $R_1$ and $R_3$ are related to $R_2 = C$ by $\pi$ rotations in $SO(3)$. This subgroup is anomaly-free because it preserves the 4th component of the $SO(5)$ vector, so the restriction of the Euler class to this subgroup is trivial and the invariant operator $M_1 + M_{-1}$ can drive the system to a trivial phase.

However, if we include the $\pi$ rotation $R_4 = \mathrm{diag}(1, 1, 1, -1, -1)$ in the $U(1)$ magnetic symmetry, there is an anomaly which can be written $\frac{1}{2} A_1 A_2 A_3 A_4 \in H^4(\mathbb{Z}_2^4, U(1))$, where $A_j$ is a gauge field coupling to the symmetry $R_j$. This anomaly tells us that the confined phases obtained by breaking $R_4$ and activating the monopole perturbation $\pm(M_1 + M_{-1})$ (preserving the remaining $\mathbb{Z}_2^3$), which are gapped nondegenerate phases, are actually distinct $\mathbb{Z}_2^3$ SPT phases differing by the class $\frac{1}{2} A_1 A_2 A_3 \in H^3(\mathbb{Z}_2^3, U(1))$. Thus $R_4$ may be considered an SPT entangler, and its $U(1)$ enhancement into the magnetic symmetry as a pivot.

If we preserve the $K = U(1) \rtimes \mathbb{Z}_2^3$ symmetry,[4] one possible symmetry relevant operator is the mass term $m^2 \sum_j |z_j|^2$. For $m^2 \gg 0$, the scalars $z_j$ can be discarded and we have a free $U(1)$ gauge field, which is in a Coulomb phase. This can is simply the superfluid phase for the $U(1)$ pivot symmetry.

For $m^2 \ll 0$, the scalars condense ($\langle z_j \rangle \neq 0$) and $(n^x, n^y, n^z)$ is an order parameter for an $O(3) = SO(3) \rtimes \mathbb{Z}_2^C$ SSB phase. In that phase, higher order anisotropies preserving $\mathbb{Z}_2^3$ become important and lock in the $(n^x, n^y, n^z)$ and we obtain the $\mathbb{Z}_2^3$ FM.

---

[4]$\mathbb{Z}_2^3$ acts on $U(1)$ by the product map $\mathbb{Z}_2^3 \to \mathbb{Z}_2 = \mathrm{Aut}(U(1))$ taking $(x, y, z) \mapsto xyz$, with $x, y, z = \pm 1$.

This matches the phase diagram in Fig. 2 if we identify the group $K$ with the lattice operators shown in Table 1. We can also deduce some of the crystalline symmetry, such as the $D_6$ (dihedral group of six elements) point group of transformations preserving a triangular plaquette. The fields transform according to the block diagonal matrices with a $3 \times 3$ permutation matrix in the upper left and identity of the lower $2 \times 2$ block. The reflection line which passes through the $A$ sublattice must fix $R_1$ and exchange $R_2$ and $R_3$. It is therefore a transposition matrix of the 2nd and 3rd columns, and similarly for the other elements. The $C_3$ rotation forbids other relevant operators in this theory which otherwise could flow to an $O(4)$-symmetric model [99] (see Sec. 4.1.2 for a case where this is expected to happen).

We remark that although the leading cubic anisotropy term $n_x^4 + n_y^4 + n_z^4$ is expected to be relevant for the $O(3)$ criticality, the difference between the critical exponent in both cases are very small [82, 95–97]. Furthermore, at the $SO(5)$ DQCP, the cubic anisotropy in our system is expected to be irrelevant. This can be argued by virtue of the known fact that adding a $C_4$ anisotropy to the $SO(5)$ DQCP is irrelevant [68, 86]. This is known in the context of e.g. Neel-VBS transition where $C_4$ lattice rotation gives rise to an emergent $U(1)$. More precisely, if we have the vector $(n_x, n_y, n_z, V_x, V_y)$, then the $C_4$ rotation traditionally acts on $(V_x, V_y)$, becoming an emergent $U(1)$. Due to the $SO(5)$ symmetry of the fixed point, we can repeat this argument for other pairs of the vector's indices. Moreover, since any term with cubic anisotropy is invariant under $C_4$, the former must be irrelevant too. As a simple example, let us consider the cubic anisotropy $n_x^4 + n_y^4 + n_z^4$. An irrelevant $C_4$-anisotropic term that arises in the Neel-VBS transition is $V_x^4 + V_y^4$. Due to $SO(5)$ symmetry, we learn that, e.g., $n_x^4 + n_y^4$, $n_y^4 + n_z^4$ and $n_x^4 + n_z^4$ are all irrelevant, and the same thus holds for their symmetrized version, which recovers our aforementioned cubic anisotropy.

## 4 Generalizations

The above example illustrated the use of pivot Hamiltonians in constructing SPT phases and uncovering interesting quantum criticality between the trivial and SPT phase. It is thus natural to ask if this can be generalized to other cases, including higher dimensions. More formally, we can ask whether it is possible to generally construct (sign-problem-free) lattice models with an exact $G$ symmetry and a $U(1)$ pivot symmetry (the latter generating a $G$-SPT).

In the above example, it turns out that the triangular lattice was particularly simple, because the $U(1)$ pivot symmetry is present at the midpoint of a direct interpolation $H_0 + H_{\text{SPT}}$. In contrast, we will now study the same pivot on the Union Jack lattice, where we pointed out in our companion work that $H_0 + H_{\text{SPT}}$ does *not* commute with the pivot [62]. Nevertheless, we show that a deformed interpolation can be constructed to restore full $U(1)$ pivot symmetry at the midpoint. This procedure naturally generalizes to higher dimensional pivots, and we provide examples of pivots creating 3D SPTs and subsystem SPTs. Exploring the transition between such phases can provide breeding ground for potentially rich quantum critical points in higher dimensions; we leave the numerical study of these models to future work.

In order to reveal the structure of the pivot symmetry, in each example, we can perform a duality transformation, mapping the operators of the Hamiltonian to dual operators where the pivot Hamiltonian is onsite. This isomorphism is often called the gauging map or generalized Kramers-Wannier duality [100–108]. We will show this explicitly for the triangular lattice, from which generalizations follow similarly.

## 4.1 KW-dual of the $\mathbb{Z}_2^3$ SPT: color code and plaquette XY model

### 4.1.1 Triangular lattice

The duality can be thought of as an isomorphism of operators, where we map each 3-body term in $H_{\text{pivot}}$ to a single Pauli $Z$. This Pauli $Z$ operator lives at the center of each triangle in the triangular lattice, which are vertices of the (dual) honeycomb lattice. Alternatively, it can be thought of as the result of gauging the $\mathbb{Z}_2^2$ symmetry of the Hamiltonian, generated by $P_A P_B$ and $P_A P_C$. Graphically,

$$\begin{matrix} Z \\ Z \quad\quad Z \end{matrix} \quad \rightarrow \quad Z \quad , \tag{18}$$

$$- \begin{matrix} Z \quad\quad Z \\ Z \end{matrix} \quad \rightarrow \quad Z \quad , \tag{19}$$

$$X \quad \rightarrow \quad \begin{matrix} X \\ X \quad\quad X \\ X \quad\quad X \\ X \end{matrix} \quad . \tag{20}$$

Note that the mapping is chosen such that the pivot has no alternating sign after the mapping. Namely, it takes the simple form

$$\tilde{H}_{\text{pivot}} = \frac{1}{8} \sum_v Z_v \,. \tag{21}$$

The duality imposes a gauge constraint on the dual Hamiltonian. Because the product of six triangles is the identity, this imposes

$$\prod_{v \in \bigcirc} Z_v = -1 \tag{22}$$

on every plaquette in the honeycomb lattice. Importantly, the staggered sign of the Baxter-Wu pivot is now encoded in the minus sign of this gauge constraint. Indeed, note that the Hamiltonian $\tilde{H}_{\text{pivot}}$ under the above constraint is still frustrated.

We choose to enforce this constraint energetically by attaching projectors $\frac{1}{2}(1 - \prod_{v \in \bigcirc} Z_v)$ to the terms in the Hamiltonian. We can now dualize the Hamiltonians $H_0$ and $H_{\text{SPT}}$. First, dualizing $H_0$ gives

$$H_0 \rightarrow \tilde{H}_0 = -\sum_{\bigcirc} \frac{1 - \prod_{v \in \bigcirc} Z_v}{2} \prod_{v \in \bigcirc} X_v \tag{23}$$

$$= -\frac{1}{2} \left[ \sum_{\bigcirc} \prod_{v \in \bigcirc} X_v + \sum_{\bigcirc} \prod_{v \in \bigcirc} Y_v \right] \,. \tag{24}$$

Thus, gauging the paramagnet Hamiltonian gives the color code on the honeycomb lattice [109, 110], whose ground state has $\mathbb{Z}_2^2$ topological order.

Next, we can obtain $\tilde{H}_{\text{SPT}}$ by evolving $\tilde{H}_0$ by $\tilde{H}_{\text{pivot}}$. Using

$$e^{-i\frac{\pi}{8}Z} X e^{i\frac{\pi}{8}Z} = \frac{X - Y}{\sqrt{2}} \,, \tag{25}$$

we obtain

$$\tilde{H}_{\text{SPT}} = e^{-i\pi\tilde{H}_{\text{pivot}}} \tilde{H}_0 e^{i\pi\tilde{H}_{\text{pivot}}} \tag{26}$$

$$= -\sum_{\bigcirc} \frac{1 - \prod_{v\in\bigcirc} Z_v}{2} \prod_{v\in\bigcirc} \frac{X_v - Y_v}{\sqrt{2}} \tag{27}$$

$$= -\frac{1}{2}\left[ \sum_{\bigcirc} \prod_{v\in\bigcirc} \frac{X_v - Y_v}{\sqrt{2}} + \sum_{\bigcirc} \prod_{v\in\bigcirc} \frac{X_v + Y_v}{\sqrt{2}} \right]. \tag{28}$$

Thus in the dual language, the two Hamiltonians realize distinct symmetry-enriched topological phases.[5]

In this dual prescription, we can give an alternative proof for $[\tilde{H}_0 + \tilde{H}_{\text{SPT}}, \tilde{H}_{\text{pivot}}] = 0$. Interestingly, substituting $X = \sigma^+ + \sigma^-$ and $Y = i(\sigma^- - \sigma^+)$, we find a curious plaquette XY model defined on the honeycomb lattice

$$\frac{1}{2}(\tilde{H}_0 + \tilde{H}_{\text{SPT}}) = -\sum_{\bigcirc} \sum_{v_i\in\bigcirc, i=1,\dots,6} \sigma^+_{v_1} \sigma^+_{v_2} \sigma^+_{v_3} \sigma^-_{v_4} \sigma^-_{v_5} \sigma^-_{v_6}, \tag{29}$$

where, for each hexagon, the Hamiltonian contains a sum of 20 terms, consisting of all the distinct ways to place three $\sigma^+$ and three $\sigma^-$ operators on the six vertices around the hexagon. These individual "ring exchange" terms commute with $\tilde{H}_{\text{pivot}}$. In this picture, it is then apparent that the model has a $U(1)$ symmetry generated by the pivot.

In fact, we can understand why this midpoint must have a $U(1)$ symmetry. First, we remark that naively, it appears that there is a mismatch between the labeling of the charges because after the mapping, the dual pivot Eq. (21) now has order eight rather than order two. This conundrum is resolved by noticing that certain charges of the pivot cannot appear at low energies since they violate the gauge constraint. For example, the operator $\mathcal{O} = \sigma^+_{v_1} \sigma^+_{v_2} \sigma^+_{v_3} \sigma^+_{v_4} \sigma^-_{v_5} \sigma^-_{v_6}$ would technically have charge $\frac{1}{4}\cdot 2 = \frac{1}{2}$ under the pivot. However, note that $\mathcal{O}$ can be rewritten as

$$\prod_{v\in\bigcirc} X_v \frac{1 - Z_{v_1}}{2} \frac{1 - Z_{v_2}}{2} \frac{1 - Z_{v_3}}{2} \frac{1 - Z_{v_4}}{2} \frac{1 + Z_{v_5}}{2} \frac{1 + Z_{v_6}}{2}. \tag{30}$$

The six projectors on the right project to a state that satisfies $\prod_{v\in\bigcirc} Z_v = 1$. Therefore, such an operator cannot appear in the constraint subspace (22). A similar statement can be made for any operator with half-integer charge. On the other hand, the operator $\mathcal{O}' = \sigma^+_{v_1} \sigma^+_{v_2} \sigma^+_{v_3} \sigma^+_{v_4} \sigma^+_{v_5} \sigma^-_{v_6}$ has unit charge under the pivot and is thus a valid perturbation in the gauge-invariant subspace. However, this term cannot appear in the expansion of $\tilde{H}_0 + \tilde{H}_{\text{SPT}}$, since this term is charge neutral under the $\mathbb{Z}_2$ pivot (in the original ungauged language). To conclude, the only allowed term that can appear in the expansion of $\frac{1}{2}(\tilde{H}_0 + \tilde{H}_{\text{SPT}})$ has to have an equal number of $\sigma^+$ and $\sigma^-$ operators. This is precisely why there is a $U(1)$ symmetry.[6]

---

[5]To see this, notice that both Hamiltonians explicitly have both time-reversal $\mathcal{T} = K$, and $\prod_v X_v$ as a global symmetry. In the case of $\tilde{H}_{\text{SPT}}$ these two symmetries enrich the color code topological order by observing that the action of the symmetry swaps the two plaquette terms. Therefore, the anyons which are violations of the plaquette terms will be permuted under the symmetry action. This permutation is easiest to express by mapping the color code to two toric codes [110–113], for which the permutation swaps the anyons between the two copies.

[6]A general criterion that guarantees the full $U(1)$ pivot symmetry can also be made for general pivots. See Appendix B of our companion work, Ref. [62].

### 4.1.2 The Union Jack lattice

Let us now consider a different 3-colorable lattice in 2D: the Union Jack lattice. The same $\mathbb{Z}_2^3$ SPT phase can also be defined on this lattice [49, 114] and can be similarly obtained by evolving $H_0$ with the pivot

$$H_{\text{pivot}} = \frac{1}{8} \sum_{a,b,c \in \Delta} (-1)^\Delta Z_a Z_b Z_c \,, \tag{31}$$

where $(-1)^\Delta$ is a sign which can be assigned in an alternating fashion to all triangles such that adjacent triangles have an opposite sign. However, unlike the triangular lattice, we find that $[H_0 + H_{\text{SPT}}, H_{\text{pivot}}] \neq 0$ [62]. Therefore, although the midpoint has a $\mathbb{Z}_2$ symmetry given by $e^{i\pi H_{\text{pivot}}}$, this $\mathbb{Z}_2$ symmetry is not enhanced to a $U(1)$ symmetry.

It is informative to see what goes wrong, which will also clarify how to modify the model to obtain a $U(1)$ pivot symmetry. This can be revealed by going to the dual variables. A similar calculation shows that $\tilde{H}_0$ is now the color code on the square-octagon lattice, and $\tilde{H}_{\text{SPT}}$ is the $\mathbb{Z}_2$ or $\mathbb{Z}_2^T$ enriched color code on the same lattice. Expanding $\tilde{H}_0 + \tilde{H}_{\text{SPT}}$ in terms of raising and lowering operators, we find that it takes the following form

$$
\begin{aligned}
\frac{1}{2}(H_0 + \tilde{H}_{\text{SPT}}) = &-\sum_{\square} \sum_{v_i \in \square} \sigma_{v_1}^+ \sigma_{v_2}^+ \sigma_{v_3}^- \sigma_{v_4}^- \\
&-\sum_{\bigcirc} \sum_{v_i \in \bigcirc} \sigma_{v_1}^+ \sigma_{v_2}^+ \sigma_{v_3}^+ \sigma_{v_4}^+ \sigma_{v_5}^- \sigma_{v_6}^- \sigma_{v_7}^- \sigma_{v_8}^- + \tilde{H}_{\text{ch}} \,,
\end{aligned}
\tag{32}
$$

where

$$\tilde{H}_{\text{ch}} = -\sum_{v_i \in \bigcirc} (\sigma_{v_1}^+ \sigma_{v_2}^+ \sigma_{v_3}^+ \sigma_{v_4}^+ \sigma_{v_5}^+ \sigma_{v_6}^+ \sigma_{v_7}^+ \sigma_{v_8}^+ + h.c.) \,. \tag{33}$$

Therefore, we see that $\frac{1}{2}(\tilde{H}_0 + \tilde{H}_{\text{SPT}})$ only fails to commute with $\tilde{H}_{\text{pivot}}$ because of $\tilde{H}_{\text{ch}}$, which is charge neutral under the $\mathbb{Z}_2$ subgroup, but contains terms of charge $\pm 2$ under the full $U(1)$ pivot. Therefore, the $\mathbb{Z}_2$ pivot symmetry at the midway of the direct interpolation is not enlarged to $U(1)$.

Nevertheless it is clear that $[\frac{1}{2}(\tilde{H}_0 + \tilde{H}_{\text{SPT}}) - \tilde{H}_{\text{ch}}, \tilde{H}_{\text{pivot}}] = 0$. Indeed, by reversing this duality, we obtain an expression for a Hamiltonian $H_{\text{ch}}$ for which $\frac{1}{2}(H_0 + H_{\text{SPT}}) - H_{\text{ch}}$ has the full $U(1)$ pivot symmetry. One can therefore construct an alternate path to interpolate between trivial and SPT phases. For example, we can consider the path

$$H(\alpha) = (1 - \alpha)H_0 + \alpha H_{\text{SPT}} - 2\sqrt{\alpha(1 - \alpha)} H_{\text{ch}} \,, \tag{34}$$

which will commute with the $U(1)$ pivot at $\alpha = 0.5$. Following the methods of Sec. 2, one can then investigate numerically whether there is an intermediate phase separating the trivial and SPT phases along this path and attempt to add terms that suppresses the intermediate phase in order to drive it to a multicritical point. The anomalous internal $U(1) \rtimes \mathbb{Z}_2^3$ symmetry of such a critical point matches the triangular lattice model we studied in detail. The $\mathbb{CP}^1$ model can match this anomaly, however, without the $C_3$ crystalline symmetry of the triangular lattice, it seems unlikely that the $SO(5)$-symmetric critical point is stable due to relevant operators that are now allowed. More likely, it will flow to an $O(4)$ DQCP (see the field theory discussion in Ref. [99]).[7]

---

[7]More precisely, following the discussion of Ref. [99]: if $(n_x, n_y, n_z, V_x, V_y)$ is our $SO(5)$ vector of the DQCP, the lattice symmetry of the Union Jack lattice allows us to separately tune, say, $n_z^2$, which leaves the remaining $O(4)$ vector $(n_x, n_y, V_x, V_y)$, which is believed to allow for a continuous DQCP transition [22, 115].

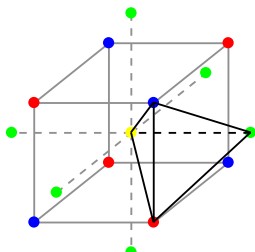

Figure 8: The BCC lattice is four-colorable. The pivot consists of an alternating sum of $ZZZZ$ terms for all tetrahedra.

To summarize, although the midpoint of a direct interpolation does not necessarily have $U(1)$ pivot symmetry, we can devise an alternate path that does. This construction naturally extends to higher dimensions, and is therefore a viable method to hunt for interesting quantum critical points. We outline two possible generalizations in 3D, where in both cases the direct interpolation does not commute with $H_{\text{pivot}}$, but an alternate path can be constructed. The corresponding dual models are described in Appendix D.

## 4.2 $\mathbb{Z}_2^4$ or $\mathbb{Z}_2^3 \times \mathbb{Z}_2^T$ SPT on the BCC lattice

Generalizing to $d$ spatial dimensions we can similarly create a $\mathbb{Z}_2^{d+1}$ SPT. Here, we consider the story in 3 dimensions. Unlike the triangular lattice in 2D, it is impossible to tile (flat) 3D space with regular tetrahedra. Therefore, we consider the 3D version of the Union Jack lattice, the 4-colorable BCC lattice, with additional edges connecting the body-centers between adjacent cubes. This is known as the tetragonal disphenoid honeycomb. The colors are given by viewing the BCC lattice as two disjoint cubic lattices and assigning two colors to each cubic lattice in a checkerboard pattern. Let us call the two sublattices within each cubic lattice $A, B$ and $C, D$, respectively as shown in Fig. 8. The pivot Hamiltonian is given by

$$H_{\text{pivot}} = \frac{1}{16} \sum_{\triangle} (-1)^{\triangle} ZZZZ, \tag{35}$$

where the sign associated with each tetrahedron can be alternated such that tetrahedra that share a face have opposite signs.[8]

Conjugating $H_0$ by $U = e^{-i\pi H_{\text{pivot}}}$, we obtain the $\mathbb{Z}_2^4$-SPT model

$$H_{\text{SPT}} = UH_0U^\dagger = -\sum_v \quad , \tag{36}$$

where each of the 24 triangles surrounding the $X$ operator is a Controlled-Controlled-$Z$ gate.[9]

As a side problem in geometry, it would be interesting to find a four-colorable lattice of tetrahedra with less than 16 tetrahedra touching each vertex. The SPT transition on such a lattice (if it exists) would inherently have a $U(1)$ pivot symmetry, without having to appeal to the symmetrization process in Sec. 4.1.2.

---

[8]More formally, choosing an ordering of colors induces a branching structure on this simplicial complex, from which an orientation for each tetrahedron can be assigned.

[9]This is defined as $CCZ_{ijk} = (-1)^{s_i s_j s_k}$ where $s = \frac{1-Z}{2}$.

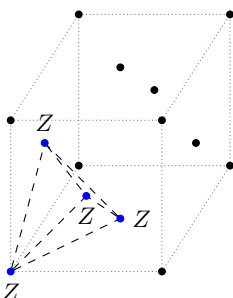

Figure 9: The tetrahedral Ising interaction on the FCC lattice is constructed for a vertex along with three adjacent face-centers within the same cube. The pivot is constructed from an alternating sum of such terms.

### 4.3 Subsystem SPT in 3D

We consider the FCC lattice and consider the pivot

$$H_{\text{pivot}} = \frac{1}{8}\sum_{\triangle}(-1)^{\triangle}ZZZZ\,, \tag{37}$$

where $\triangle$ denotes tetrahedra built from a vertex along with three adjacent face-centers within the same cube as shown in Figure 9.

Pivoting the trivial Hamiltonian $H_0$ results in [116]

$$H_{\text{SPT}} = -\sum_{v} \quad X_{v} \quad, \tag{38}$$

where the 24 blue edges denote $CZ$ operators connecting the twelve face-centers surrounding $X$. This SPT is protected by a combination of time-reversal symmetry and planar subsystem symmetries defined as flipping spins along individual (100), (010), and (001) planes of the FCC lattice.

As we have mentioned above, both these three-dimensional SPT models do not give rise to a $U(1)$ pivot symmetry for the direct interpolation $H_0 + H_{\text{SPT}}$. However, using the symmetrization procedure introduced in Sec. 4.1.2, we can obtain a lattice model which has both the symmetry necessary for the SPT phase as well as the $U(1)$ pivot symmetry, making them very interesting candidates for exotic topological criticality.

## 5 Outlook

Using the idea of pivots, we have arrived at a model on the triangular lattice which has a non-onsite $U(1)$ pivot symmetry in addition to the onsite $\mathbb{Z}_2^3$ symmetry protecting the nearby SPT phase. Using a Monte Carlo and field-theoretic analysis, we have argued that this model supports an SPT multicriticality described by $SO(5)$ DQCP. The stability of the latter crucially relies on the aforementioned $U(1)$ pivot symmetry. We have moreover described how one can more generally construct sign-problem-free lattice models with such an anomalous symmetry group, which can aid the search for interesting topological criticality.

We remark that even if a direct interpolation for some $G$-SPT, i.e., $H_0 + H_{\text{SPT}}$, does not enjoy a $U(1)$ pivot symmetry, then one can always look at the symmetrized Hamiltonian[10]

$$\frac{1}{2\pi}\int_0^{2\pi} e^{-i\alpha H_{\text{pivot}}}(H_0 + H_{\text{SPT}})e^{i\alpha H_{\text{pivot}}}\text{d}\alpha. \tag{39}$$

Indeed, at least in the case of $H_0 = -\sum X$ and diagonal pivots, one can straightforwardly show that Eq. (39) commutes with both $H_{\text{pivot}}$ and $G$. However, while this formula seems conceptually appealing, there are two concerns: (1) will this expression be nonzero? (2) And how does one compute it? Indeed, directly calculating Eq. (39) is very challenging due to the non-onsite nature of the pivot Hamiltonian.

Essentially, the procedure outlined in Section 4.1.2 gives a constructive way of obtaining Eq. (39) through means of the Kramers-Wannier transformation. We have shown for a variety of examples (in 2D and 3D) that the result of this computation indeed gives a nonzero Hamiltonian for Eq. (39). We note that these are all manifestly sign-problem-free. It would be very interesting to numerically study these resulting models, in search for exotic topological criticality. One of the examples we gave in Sec. 4 is for a subsystem SPT (SSPT) phase. We note that other SSPTs in 3D could also be considered [108, 117–120] that can potentially give rise to stable phase transitions with interesting renormalization properties beyond ordinary CFTs [121–124].

In fact, even for the triangular lattice model which we studied in detail in the present work, it would be interesting to directly simulate the dual honeycomb plaquette XY model Eq. (29). Multibranch cluster updates [125] might speed up the simulations, allowing to study larger system sizes, and compare critical exponents obtained from other models described by $SO(5)$ DQCP. Perhaps one can even directly observe the emergent $SO(5)$ symmetry.

Let us also note that the direct transition between trivial and SPT phases studied in Sec. 2 implies the direct transition of various topological orders via gauging the various subgroups of $\mathbb{Z}_2^3$ (we give an extended list in Appendix C). For example, gauging the diagonal $\mathbb{Z}_2$ symmetry gives a direct transition between the toric code and double semion topological orders (where in fact, the $U(1)$ becomes non-local). It would be interesting to work out the physical interpretation of these multicritical points between topologically ordered phases.

# Acknowledgments

NT is supported by NSERC. AV and RV are supported by the Simons Collaboration on Ultra-Quantum Matter, which is a grant from the Simons Foundation (651440, A.V.). RV is supported by the Harvard Quantum Initiative Postdoctoral Fellowship in Science and Engineering.

# A   Details of the cluster algorithm

In this Appendix, we present a variant of the cluster algorithm [80] used to simulate the Hamiltonian (6).

We begin by reviewing the Stochastic Series Expansion [78, 79]. Given a Hamiltonian of the form

$$H_{\text{QMC}} = -\sum_{t,a} H_{t,a}, \tag{40}$$

---

[10]Since $H_{\text{pivot}}$ is a sum of local commuting terms, such a symmetrized Hamiltonian is always local.

where $t$ and $a$ are indices to denote they type and position of each term in the Hamiltonian, the partition function at inverse temperature $\beta$ can be written as

$$Z = \sum_{\{\sigma\}} \sum_{S_M} \frac{\beta^n (M-n)!}{M!} \langle\sigma| \prod_{p=1}^{M} H_{t,a} |\sigma\rangle \,, \tag{41}$$

where $|\sigma\rangle$ is a basis of states, and $S_M$ is a sequence of the indices $t, a$ called the "operator-string", which denotes all the possible insertions of operators $H_{t,a}$ into the expectation value. The weights in the partition function can be considered as that of a classical system if all the entries of $H_{t,a}$ in the basis of $\sigma$ are non-negative.

The Hamiltonian we will be simulating consists of three types of operators $t = 0, 1, 2$

$$H_{0,v} = h \,, \tag{42}$$

$$H_{1,v} = hX_v(1 + k\mathcal{O}_v^{\text{ring}}) \,, \tag{43}$$

$$H_{2,b} = |J|(1 - \text{sgn}(J)Z_{b_1}Z_{b_2}) \,, \tag{44}$$

where

$$\mathcal{O}_v^{\text{ring}} = \quad \begin{array}{c}\text{CZ}\end{array} \quad . \tag{45}$$



Here, $H_{0,v}$ and $H_{1,v}$ are defined for each site of the triangular lattice, and $H_{2,b}$ is given for each bond of the red, blue, or green sublattice.

For the choices above, $H_{\text{QMC}}$ can be related to the original Hamiltonian (6) via a constant shift

$$H_{\text{QMC}} = H - (h + 3|J|)N_v \,, \tag{46}$$

provided that we relate $h$ and $k$ to $\alpha$ via

$$h = 1 - \alpha \,, \qquad\qquad k = \frac{\alpha}{1 - \alpha} \,. \tag{47}$$

Here, $N_v = L^2$ is the number of vertices and the factor of 3 comes from the fact that the number of bonds on the triangular lattice (with periodic boundary conditions) is equal to $3N_v$. The operators $H_{t,a}$ all have positive entries in the $Z$ basis for $-1 \leq k \leq 1$, corresponding to $\alpha \leq 0.5$. Nevertheless, the phase diagram for $\alpha > 0.5$ can be obtained by using the $\mathbb{Z}_2$ duality (7), which maps $\alpha \to 1 - \alpha$.

The only change to the cluster algorithm is the calculation of the probability of flipping clusters in the off-diagonal update. (The diagonal update remains the same, and consists of inserting or removing $H_{0,v}$ or $H_{2,b}$.)

The off-diagonal update consists of flipping all propagated spins within a cluster and swapping $H_{0,v}$ and $H_{1,v}$ at all the end points. In the original cluster update (i.e., when $k = 0$), since the two operators have equal weights $h$, such a flip is always accepted (in practice we choose the probability $p = 1/2$ to ensure ergodicity). In general, we need to calculate the ratio of the weights when $k \neq 0$.

First, consider the endpoints of a cluster $C$. If the initial operator at the end point is $H_{0,v}$, then the weights before and after are respectively $h$ and $h(1 + k\mathcal{O}_{\text{af}}^{\text{ring}})$, where $\mathcal{O}_{\text{af}}^{\text{ring}}$ denotes the evaluation of $\mathcal{O}^{\text{ring}}$ *after* the flipping all propagated spins within the cluster. Similarly, if the initial operator at the end point is $H_{1,v}$, the weights before and after are respectively $h(1 + k\mathcal{O}_{\text{be}}^{\text{ring}})$ and $h$, where $\mathcal{O}_{\text{be}}^{\text{ring}}$ denotes the evaluation of *before* the flip.

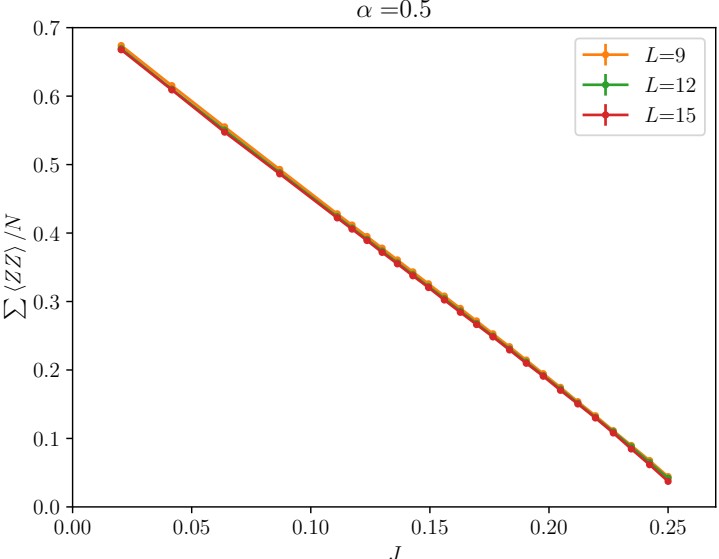

Figure 10: We observe no visible discontinuity in $\frac{dE}{dJ}$, excluding the possibility of a conventional first-order transition.

In addition, there can also be $H_{1,\nu}$ operators that are not end points of the cluster $C$ to be flipped, but where the support of cluster overlaps with $\mathcal{O}_\nu$, thus changing the weight from $h(1 + k\mathcal{O}_{\text{be}}^{\text{ring}})$ to $h(1 + k\mathcal{O}_{\text{af}}^{\text{ring}})$. To conclude, the ratio of the weights in the partition function before and after the flip is

$$\frac{W_{\text{af}}}{W_{\text{be}}} = \frac{\prod_{(0,\nu)\in C}(1 + k\mathcal{O}_{\text{af}}^{\text{ring}})}{\prod_{(1,\nu)\in C}(1 + k\mathcal{O}_{\text{be}}^{\text{ring}})} \prod_{(1,\nu)\notin C} \frac{1 + k\mathcal{O}_{\text{af}}^{\text{ring}}}{1 + k\mathcal{O}_{\text{be}}^{\text{ring}}}. \tag{48}$$

Therefore, using the Metropolis update we can choose to flip these clusters with probability

$$p = \min\left(\frac{W_{\text{af}}}{W_{\text{be}}}, 1\right). \tag{49}$$

Note that the above choice is not well-defined for $k = 1$ (i.e. $\alpha = 0.5$ where the $U(1)$ pivot symmetry is present.) This is because $(1 + k\mathcal{O}^{\text{ring}})$ will take values 0 or 2, and so we potentially run into dividing by zero. To avoid this, we take the probability in this case to be

$$p = \min\left(\lim_{k\to 1^-} \frac{W_{\text{af}}}{W_{\text{be}}}, 1\right). \tag{50}$$

In practice, we can evaluate this by keeping track of the number of times we encounter a zero when evaluating $(1 + kO^{\text{ring}})$. The probability is then chosen to be 0 if more zeros are encountered in the numerator than in the denominator, and 1 vice versa. In the case that they appear equally, $\frac{W_{\text{af}}}{W_{\text{be}}}$ is evaluated after removing all factors $(1 + k\mathcal{O}^{\text{ring}})$ that are zero in the product.

## B  More numerical results

We provide evidence that the transition between the FM and SF phases at $\alpha = 0.5$ is continuous. Note that from the Hellman-Feynman theorem, $\frac{dE}{dJ} = \langle H_{\text{NNN}} \rangle = \langle \sum ZZ \rangle$, and since the

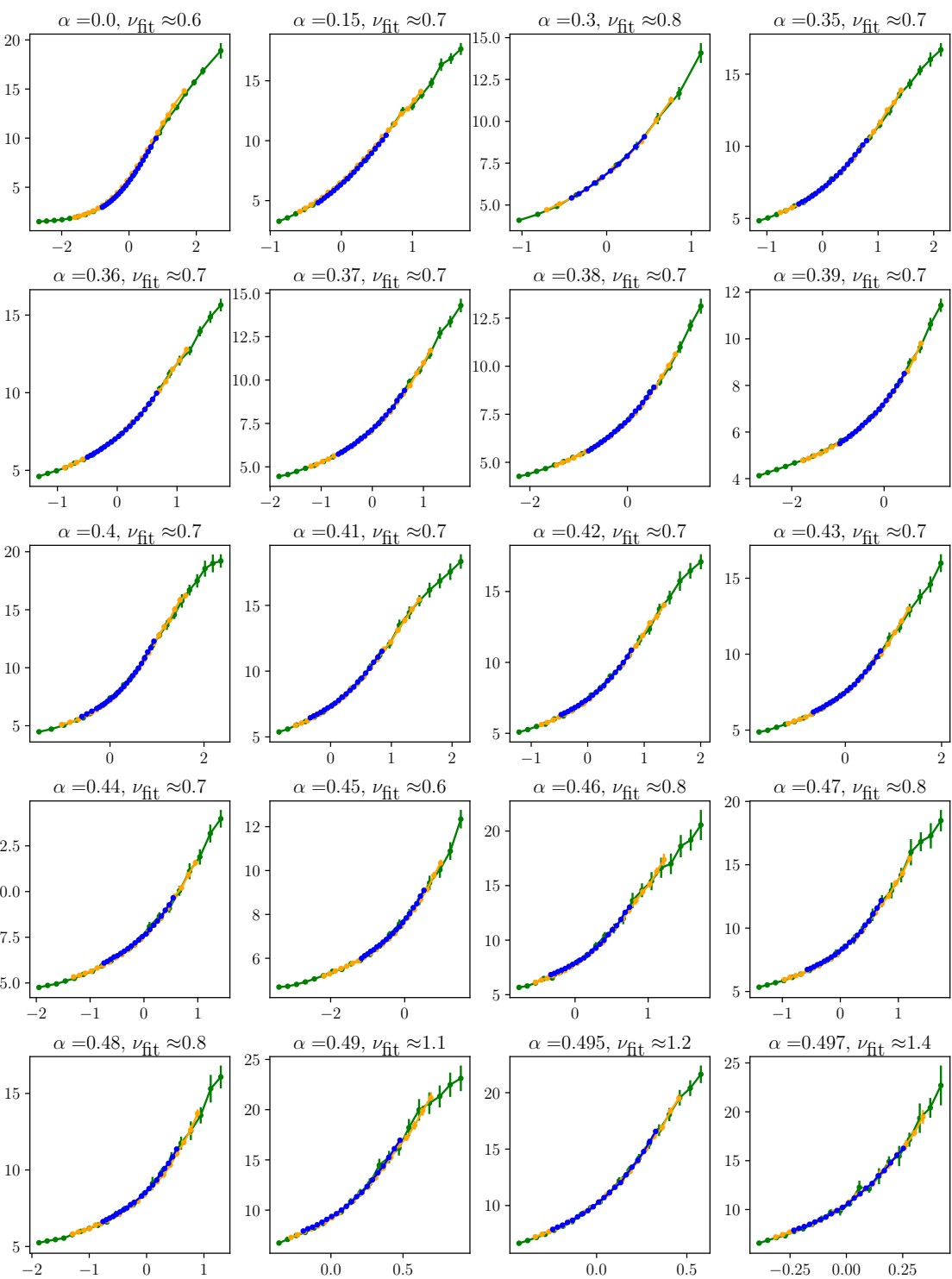

Figure 11: Binder ratio collapse. For each subplot, the Binder ratio $B$ is plotted against $(J - J_c)L^{1/\nu_{\text{fit}}}$. The best fit values of $\nu_{\text{fit}}$ for each value of $\alpha$ is shown and the value of $J_c$ are used to plot the phase boundary (black line in Fig. 2).

Table 2: Summary of various direct transitions between topological phases obtained by gauging the DQCP transition between trivial and $\mathbb{Z}_2^3$ SPT. For each row in the table, the multicritical point is found at $\frac{1}{2}(H_0 + H_2) + J_c H_J$ where $J_c \approx 0.21$. The color coding of the table is for ease of comparison to the phase diagram in Fig. 2.

| $H_0$ | $H_{\text{SPT}}$ | $H_0 + H_{\text{SPT}}$ | $H_{\text{pivot}}$ |
|---|---|---|---|
| $\mathbb{Z}_2$ TC | $\mathbb{Z}_2$ TC with $\mathbb{Z}_2^2$ fractionalization | $\mathbb{Z}_2^2$ FM | Non-local |
| $\mathbb{Z}_2$ TC | DS | FM | Non-local |
| $\mathbb{Z}_2$ TC$^2$ | $\mathbb{Z}_2$ TC$^2$ with $\mathbb{Z}_2$ or $\mathbb{Z}_2^T$ permutation | $\mathbb{Z}_2$ AFM | $Z$ |
| $\mathbb{Z}_2$ TC$^2$ | $\mathbb{Z}_4$ TC | Confined | Non-local |
| $\mathbb{Z}_2^3$ TC | Twisted $\mathbb{Z}_2^3$ gauge theory | Confined | Non-local |

Ising term appears in the stochastic series expansion, its expectation value can be computed via

$$\frac{\sum \langle ZZ \rangle}{N} = -\frac{\langle n_{ZZ} \rangle}{\beta N J} + 3 \operatorname{sgn}(J), \tag{51}$$

where $n_{ZZ}$ is the number of Ising operators that appears in the operator string of the stochastic series expansion.

In Fig. 10, we plot $\sum \langle ZZ \rangle / N$ as a function of $J$. We observe a continuous function, which excludes the possibility of a first order transition.

Fig. 11 shows the finite size scaling of the Binder ratio for the critical point between trivial and FM phases for various values of $\alpha$. For $\alpha = 0$, we obtain $\nu_{\text{fit}} \approx 0.6$, in agreement with the Ising criticality. For $0 < \alpha < 0.47$ the best approximate of $\nu_{\text{fit}}$ is found to be within the range $[0.6, 0.8]$, in agreement with $O(3)$ criticality [95].

# C  Other direct transitions in 2D

The $G = \mathbb{Z}_2^3$ SPT can be described by a cocycle $\frac{1}{2}A_1 A_2 A_3 \in H^3(G, \mathbb{R}/\mathbb{Z})$. We can study what happens when we gauge the various subgroups of $\mathcal{G}_U$ (we only mention the cases where the SPT is not trivialized after restricting to that subgroup). We summarize the resulting gauged models in Table 2.

1. Gauging one $\mathbb{Z}_2$. In this case, the SPT gauges to a toric code, with $\mathbb{Z}_2^2$ symmetry fractionalization on the $m$ anyon given by the cocycle $A_2 A_3 \in H^2(\mathbb{Z}_2^2, \mathbb{Z}_2)$. This can be seen as the projective representation of the cluster state that is decorated on the $m$-string.

2. Gauging the diagonal $\mathbb{Z}_2$ subgroup. The response reduces to $\frac{1}{2}A^3 \in H^3(\mathbb{Z}_2, \mathbb{R}/\mathbb{Z})$. Thus the SPT gauges to the Double Semion topological order (the $\mathbb{Z}_2$ twisted Dijkgraaf-Witten gauge theory).

3. Gauging the A and B $\mathbb{Z}_2^2$ subgroup. Gauging the SPT gives a $\mathbb{Z}_2^2$ topological order where the two toric codes are swapped under the global $\mathbb{Z}_2$ global symmetry. Note that the same happens if we instead gauge the AB and BC subgroups because under the transformation

$$A_1 \rightarrow A_1, \tag{52}$$
$$A_2 \rightarrow A_2, \tag{53}$$
$$A_3 \rightarrow A_1 + A_2 + A_3. \tag{54}$$

The response is given by

$$\frac{1}{2}(A_1 A_2 A_3 + A_1 A_2^2 + A_2 A_1^2) \tag{55}$$

$$= \frac{1}{2}A_1 A_2 A_3 + \frac{1}{4}(A_1 dA_2 + A_2 dA_1). \tag{56}$$

Since the last two terms is exact, the response is invariant under such a transformation.

4. Gauging the AB and BC $\mathbb{Z}_2^2$ subgroup with time-reversal symmetry. We can do a similar transformation as above but set the field $A_3$ to zero afterwards

$$A_1 \to A_1, \tag{57}$$

$$A_2 \to A_2, \tag{58}$$

$$A_3 \to A_1 + A_2. \tag{59}$$

The form $\frac{1}{4}(A_1 dA_2 + A_2 dA_1)$ is exact on an orientable spacetime manifold. However, if the manifold is non-orientable, then the integration by parts gives a 1+1D SPT $\frac{1}{2}A_1 A_2$ "decorated" on the orientation reversing 2-cycle, which is Poincare dual to the first Stiefel-Whitney class $w_1$. Therefore, the response is given by $\frac{1}{2}w_1 A_1 A_2$. After gauging, the time-reversal symmetry swaps the two copies of the toric code.

5. Gauging the $A$ and $BC$ $\mathbb{Z}_2^2$ subgroup. The response is $\frac{1}{2}A_1 A_2^2$, and the SPT gauges to a twisted $\mathbb{Z}_2^2$ topological order, in the same phase as the $\mathbb{Z}_4$ toric code.

6. Gauging the full $\mathbb{Z}_2^3$ group. This case is known to be equivalent to the topological order of the group $D_8$ (the dihedral group of eight elements).

# D  Generalized XY models in 3D

## D.1  Dual of the 3D SPT: color code topological order

Let us gauge the $\mathbb{Z}_2^3$ subgroup generated by AB, AC, and AD. The dual Hilbert space is defined on each tetrahedron of the BCC lattice, which is equivalent to the vertices of the bitruncated cubic honeycomb. Each cell of the honeycomb consists of six square faces and eight hexagonal faces. The pivot Hamiltonian takes the form

$$\tilde{H}_{\text{pivot}} = \frac{1}{16}\sum_v Z_v. \tag{60}$$

The duality imposes the following gauge constraint on the dual Hamiltonian.

$$\prod_{v \in \square} Z_v = 1, \tag{61}$$

$$\prod_{v \in \bigcirc} Z_v = -1, \tag{62}$$

where $\square$ and $\bigcirc$ are square and hexagon faces of the lattice. This can be enforced by attaching projectors to each term in the Hamiltonian.

First consider the dual of $H_0$. We obtain

$$\tilde{H}_0 = -\sum_b \prod_{v \in b} X_v \mathcal{P}_b, \tag{63}$$

where $b$ denotes each bitruncated cube and $\mathcal{P}_b$ enforces the constraint on all 14 faces of $b$. The ground state is that of the 3D color code [126], which is in the same topological order as three toric codes.

Let us define

$$\tilde{X} = e^{-i\frac{\pi}{16}Z} X e^{i\frac{\pi}{16}Z} = \cos\left(\frac{\pi}{8}\right) X - \sin\left(\frac{\pi}{8}\right) Y, \tag{64}$$

we obtain

$$\tilde{H}_{\text{SPT}} = e^{-i\pi\tilde{H}_{\text{pivot}}} \tilde{H}_0 e^{i\pi\tilde{H}_{\text{pivot}}} \tag{65}$$

$$= -\sum_b \prod_{v\in b} \tilde{X} \mathcal{P}_b. \tag{66}$$

The Hamiltonian has $\mathcal{T} = K$ and $\prod_v X_v$ as global symmetries, which enrich the color code topological order. We can see this by how the excitations are permuted. This is most easily explained by mapping the color code to three toric codes. Then the flux loops of one species get permuted as

$$m_1 \to m_1 s_{23}, \qquad m_2 \to m_2 s_{31}, \qquad m_3 \to m_3 s_{12}, \tag{67}$$

where $s_{ij}$ are defects created by gauging the 1D cluster state corresponding to a $\mathbb{Z}_2^{(i)} \times \mathbb{Z}_2^{(j)}$ SPT phase [110, 127, 128].[11]

At the midpoint $\tilde{H}_0 + \tilde{H}_{\text{SPT}}$, we find that the Hamiltonian consists of all possible $U(1)$ symmetric terms (consisting of twelve $\sigma^+$ and twelve $\sigma^-$ operators), but also contains terms containing twenty $\sigma^+$ and four $\sigma^-$ operators (and their conjugates). These are charged $\frac{1}{8} \times \pm 16 = \pm 2$ operators, which spoil the $U(1)$ pivot symmetry.

## D.2 Dual of the 3D SSPT: Checkerboard fracton order

To see that the pivot is not enhanced to $U(1)$, we go to the dual variables by gauging the subsystem symmetry. The dual qubits live on the vertices of the cubic lattice and the pivot takes the form

$$\tilde{H}_{\text{pivot}} = \frac{1}{8} \sum_v Z_v. \tag{68}$$

Furthermore, the Hamiltonians of interest gauge to

$$H_0 \to \tilde{H}_0 = -\frac{1}{2}\left[\sum_c \prod_{v\in c} X_v + \sum_c \prod_{v\in c} Y_v\right], \tag{69}$$

$$H_{\text{SPT}} \to \tilde{H}_{\text{SPT}} = -\frac{1}{2}\left[\sum_c \prod_{v\in c} \frac{X_v - Y_v}{\sqrt{2}} + \sum_c \prod_{v\in c} \frac{X_v + Y_v}{\sqrt{2}}\right], \tag{70}$$

where $c$ denotes cubes over a checkerboard pattern of the cubic lattice. The first two models describe the checkerboard model [101] and its symmetry-enriched version, where charge and flux-type fractons get permuted under the action of time-reversal [116].

We identify the term that is charged-2 under the pivot as

$$H_{\text{ch}} \to \tilde{H}_{\text{ch}} = \sum_c \sum_{v_i\in c} \sigma^+_{v_1}\sigma^+_{v_2}\cdots\sigma^+_{v_8} + h.c. \tag{71}$$

---

[11]To see this explicitly, the loop excitation $m_{AB}$ is created by applying $\tilde{X}$ along $AB$ faces that lie within a membrane. Under time-reversal or $\prod_v X_v$ we find that $\tilde{X} \to \tilde{X}^\dagger \sim \tilde{X}S$. The application of $S$ within a membrane $AB$ is exactly what creates the defect $s_{CD}$.

Subtracting this term, we find

$$\frac{1}{2}(\tilde{H}_0 + \tilde{H}_{\text{SPT}}) - \tilde{H}_{\text{ch}} = \sum_c \sum_{\nu_i \in c} \sigma^+_{\nu_1} \sigma^+_{\nu_2} \sigma^+_{\nu_3} \sigma^+_{\nu_4} \sigma^-_{\nu_5} \sigma^-_{\nu_6} \sigma^-_{\nu_7} \sigma^-_{\nu_8}, \tag{72}$$

where each cube consists of a sum of 70 terms consisting of all the distinct ways to place four $\sigma^+$ and four $\sigma^-$ operators around each cube. These terms commute with the pivot (68). This Hamiltonian is an interesting "checkerboard" XY model defined on the checkerboard lattice, which has a pivot $U(1)$ symmetry.

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
