# Peer review of "Building models of topological quantum criticality from pivot Hamiltonians"

_SciPost Physics, doi:SciPost Phys. 14, 013 (2023)_

## Round 1 · Referee Report · Anonymous (Referee 1) · 2022-6-2

Report

This work utilizes a recently introduced notion of the pivot Hamiltonian to construct a 2+1d spin model where a direct transition between a trivial phase and an SPT is found. The pivot Hamiltonian serves as the generator of a non-onsite U(1) pivot symmetry that is respected midway between the trivial and the SPT phases and helps to stabilize a direct transition between them. The spin model constructed in this work is free of the sign problem. Monte Carlo simulations of this spin model show an interesting phase diagram. In particular, evidence has been found for the multi-critical point in the phase diagram to be described by the SO(5) deconfined quantum critical point (DQCP). This work also illustrates how the U(1) pivot symmetry can be generally used in the construction of sign-problem-free spin models of SPT transitions for various lattices and dimensions.

I think this work is technically sound and conceptually novel. The 2+1d spin model constructed using the pivot Hamiltonian has a rich phase diagram, of which the appearance SO(5) DQCP is a highly non-trivial feature. Moreover, the general idea of using the U(1) pivot symmetry to construct sign-problem-free spin models of SPT transitions is an important development in the field of quantum criticality. I believe this work deserves the publication on SciPost.

I do hope the authors can address the following questions:
1. In the last paragraph on page 10, the authors extracted the critical exponent \nu from the Binder-ratio calculation and found it consistent with the O(3) universality class. However, under Eq. (12), the authors cited [81], which claimed that the cubic anisotropy is a relevant perturbation to the O(3) universality class. Could the authors elaborate more on the universality class of the trivial-to-FM transition away from the SO(5) DQCP? Also, could the authors comment on whether the anisotropy on the O(3) critical line is important as we approach the SO(5) DQCP?

2. In Eq. (39), the authors proposed a symmetrized Hamiltonian that respects the U(1) pivot symmetry. Can the author comment on whether such a symmetrized Hamiltonian is always a sum of local terms?

  • validity: -
  • significance: -
  • originality: -
  • clarity: -
  • formatting: -
  • grammar: -

Author:  Nathanan Tantivasadakarn  on 2022-10-15  [id 2925]

(in reply to Report 1 on 2022-06-02)

  1. We thank the referee for these astute observations. Indeed, the cubic anisotropy is a relevant perturbation for the particular case of $O(3)$ criticality. However, the difference between the critical exponent $\nu$ in both cases are very small; we have added a comment in the manuscript to emphasize this. Moreover, we have added a new paragraph at the end of Section 3, which gives an argument that the cubic anisotropy is expected to be irrelevant at the $SO(5)$ DQCP.

  2. Since $H_\text{piv}$ is a sum of local commuting terms, the conjugated term is local. Moreover, the integral cannot change the locality of the integrand, such symmetrized Hamiltonian is local. We have added a footnote to clarify this.

---

## Round 1 · Referee Report · Anonymous (Referee 2) · 2022-6-12

Report

The authors present the study of a phase diagram obtained by interpolating between a trivial and a non-trivial Z_2^3 SPT. Based on related work by the same authors, they show that a microscopic U(1) symmetry occurs at the mid-way point, which provides a good starting point for a field theory analysis. Whereas a discrete Z_2 duality is always expected at the mid-way point and was already pointed out earlier, a U(1) symmetry is not generic and therefore noteworthy.
Remarkably, they find good evidence for a SO(5) QCP multicritical point at which the two SPT phases and a ferromagnet touch.
Since this SO(5) deconfined QCP has been under intense scrutiny in a variety of other models, most famously for Neel to VBS transition, finding a possible new instance of this DQCP in the context of SPTs is very interesting and brings new light on the topic.
A caveat is that the actual evidence for an SO(5) point given in the paper is not very direct: the system sizes studied in this paper do not allow for an accurate calculation of critical exponents, which are notorious for having a strong finite size drift on top of that. However, the field theory and symmetry arguments given to relate the microscopic symmetries to various SO(5) components are convincing and definitely motivate further work on this model.
I strongly recommend publication of this work.

Requested changes

A few minor things:

1- Fig 6: legend is not clear, there are more curves than legend entries?
2- When discussing Union Jack Lattice, it would be good to cite prior work on the SPT defined on that lattice:
arXiv:1508.02695
https://arxiv.org/abs/1705.01557

3- The sentence “To conclude, the only allowed term that can appear in the expansion of 1 2 (H˜ 0+ H˜ SPT).” seems to be missing a word?

  • validity: -
  • significance: -
  • originality: -
  • clarity: -
  • formatting: -
  • grammar: -

Author:  Nathanan Tantivasadakarn  on 2022-10-15  [id 2926]

(in reply to Report 2 on 2022-06-12)

  1. We thank the referee for pointing out this potential ambiguity. The dash line denotes the slope obtained by fitting the data (solid lines) with a cubic polynomial. We have added a clarification in the caption.
  2. We have added the above references
  3. We have fixed this to "To conclude, the only allowed term that can appear in the expansion of $\frac{1}{2}(\tilde H_0 + \tilde H_\text{SPT})$ has to have an equal number of $\sigma^+$ and $\sigma^-$ operators"

---

## Round 1 · Referee Report · Anonymous (Referee 3) · 2022-7-13

Report

The manuscript studies an interesting model realizing a phase transition between trivial and SPT phases. A key feature is that along the line alpha=0.5 in the phase diagram the model has a non-onsite U(1) symmetry generated by a “pivot” Hamiltonian.

The phase diagram is studied with Monte Carlo and found to show a version of deconfined criticality. This is explained by a symmetry and anomaly analysis of an effective field theory. Generalizations of the model are presented.

The paper is well-written. Numerical results and the logic behind the analysis/interpretation are described precisely. The symmetry and field theory discussion is very clear, modulo the queries below. The authors convincingly make the case, via the examples discussed here, that pivot Hamiltonians are a fruitful place for exploring novel quantum phase transitions.

I have no hesitation in recommending this paper for Scipost. I have only minor comments:

Sec 3: clarify the discussion of what symmetry is required to fix the phase diagram. Page 11 states that with U(1) and Z_2^3 symmetry there is “one other symmetry relevant operator” but as noted later in the section, the lattice symmetry which permutes the Z2 order parameters is necessary to prevent other relevant operators and so important for the structure of the phase diagram.

I did not understand the statement that breaking the lattice symmetry would give a flow to the O(4) model. If only U(1) and Z2^3 are retained then I would have thought there was no reason to expect even O(4), since in the sigma model description there can be four separate mass-like couplings for the three Z2 order parameters and the U(1) order parameter.

Page 8 “This suggests that the transition remains continuous as we approach α = 0.5.” Clarify that “This suggests” refers to the numerical data?

Eqn (29) - notation with v_i in sum - v_i does not appear elsewhere in the formula

Around Eqn (4) - maybe useful to state that U(1) is only present at one point on the interpolating line.

Since this symmetry appears accidental/fine-tuned here, can the authors comment on any settings where it could be in some sense more robust?

Page 7 “In contrast, along the self-dual line (α = 0.5), we observe that the FM order parameter decays very slowly. This is consistent with the singular cusp behavior in our phase diagram, making a precise determination of the critical point more challenging.“ What is the logic for relating these two things and is it correct? The shape of the cusp is to do with the scaling dimension of the alpha perturbation. But at alpha=0.5 this perturbation vanishes.

  • validity: -
  • significance: -
  • originality: -
  • clarity: -
  • formatting: -
  • grammar: -

Author:  Nathanan Tantivasadakarn  on 2022-10-15  [id 2927]

(in reply to Report 3 on 2022-07-13)

We thank the referee for their thorough review of our manuscript 1.We have revised the sentence to read "one possible symmetry relevant operator is the mass term". Then, the crystalline symmetry which we later identify forbids other relevant operators. 2. The argument for $O(4)$ emergent symmetry assumes starting at the $SO(5)$ DQCP point, and breaking the lattice symmetry. Starting from the $SO(5)$ DQCP with $SO(5)$ vector $(n_x,n_y,n_z,V_x,V_y)$, the usual transition is driven by tuning e.g. $a(n_x^2+n_y^2+n_z^2)$. If we break the crystalline symmetry down to a $\mathbb Z_2$ subgroup (for example, by instead using the union jack lattice), which acts by swapping the first two components, it now becomes possible to separately tune $n_x^2+n_y^2$ and $n_z^2$. Following the discussion in Ref. 99, the $n_z^2$ perturbation allows us to flow to the $O(4)$ DQCP (see also Refs. 22,115). We have updated the discussion in the main text. The above argument, which starts from the SO(5) point and breaks down to the lattice symmetries we expect to have on the union jack lattice, suggests that the O(4) DQCP can arise by tuning only two lattice parameters. Moreover, since one of these parameters tunes the SPT transition, we expect to only require one tuning parameter if we enforce the $\mathbb Z_2$ SPT entangler. 3. This has been added 4. We have added a clarification in Eq. 29 that $i= 1,\ldots,6$ 5. We have added a clarification "While it is obvious that at $H_0 + H_\text{SPT}$ (and only at this point in the interpolation between these two Hamiltonians) has a $\mathbb Z_2$ symmetry..." 6. In our companion paper: https://scipost.org/submissions/scipost_202204_00017v2/ we discuss a transition between a trivial and non-trivial SPT phase in 2+1D protected by time-reversal symmetry and a $\mathbb Z_2$ 1-form symmetry. One transition can be described by a gauged $O(2)/\mathbb Z_2$ Wilson-Fisher fixed point where the continuous $U(1)$ symmetry is emergent in a stable way. 7. They are indeed separate points. We have edited the caption of the figure to avoid this confusion. The caption now reads "In contrast, the order parameter decays very slowly for $\alpha=0.5$, meaning that it has a small value over a wider region of parameter space, making it more challenging to precisely locate the critical point. This region with a small magnetic moment is consistent with the phase boundary in Fig.2 displaying a narrow ordered region upon approaching the multicritical point. In fact, the latter seems to end in a cusp, which is expected from the scaling dimensions of $SO(5)$ deconfined criticality."

---

## Round 2 · List of Changes

• Clarified that exponents of "O(3) criticality" and "O(3) criticality with cubic anisotropy" are very close
  • Added argument that the cubic anisotropy is irrelevant at the $SO(5)$ DQCP.
  • Added footnote in outlook to explain that our symmetrized Hamiltonian is local.
  • We have fixed certain typos pointed out by the referees.

---

## Editorial Decision

published